# Multi-Scale-Denoising Residual Convolutional Network for Retinal Disease Classification Using OCT

**DOI:** 10.3390/s24010150

**Published:** 2023-12-27

**Authors:** Jinbo Peng, Jinling Lu, Junjie Zhuo, Pengcheng Li

**Affiliations:** 1State Key Laboratory of Digital Medical Engineering, School of Biomedical Engineering, Hainan University, Haiko 570228, China; jinbopeng@hainanu.edu.cn (J.P.); lujinling@mail.hust.edu.cn (J.L.); 2Key Laboratory of Biomedical Engineering of Hainan Province, One Health Institute, Hainan University, Haiko 570228, China; 3Research Unit of Multimodal Cross Scale Neural Signal Detection and Imaging, Chinese Academy of Medical Science, HUST-Suzhou Institute for Brainsmatics, Jiangsu Industrial Technology Research Institute (JITRI), Suzhou 215100, China; 4Britton Chance Center for Biomedical Photonics and MoE Key Laboratory for Biomedical Photonics, Wuhan National Laboratory for Optoelectronics, Huazhong University of Science and Technology, Wuhan 430074, China

**Keywords:** convolutional neural network, retinal disease classification, optical coherence tomography (OCT), multi-scale-denoising residual convolutional network

## Abstract

Macular pathologies can cause significant vision loss. Optical coherence tomography (OCT) images of the retina can assist ophthalmologists in diagnosing macular diseases. Traditional deep learning networks for retinal disease classification cannot extract discriminative features under strong noise conditions in OCT images. To address this issue, we propose a multi-scale-denoising residual convolutional network (MS-DRCN) for classifying retinal diseases. Specifically, the MS-DRCN includes a soft-denoising block (SDB), a multi-scale context block (MCB), and a feature fusion block (FFB). The SDB can determine the threshold for soft thresholding automatically, which removes speckle noise features efficiently. The MCB is designed to capture multi-scale context information and strengthen extracted features. The FFB is dedicated to integrating high-resolution and low-resolution features to precisely identify variable lesion areas. Our approach achieved classification accuracies of 96.4% and 96.5% on the OCT2017 and OCT-C4 public datasets, respectively, outperforming other classification methods. To evaluate the robustness of our method, we introduced Gaussian noise and speckle noise with varying PSNRs into the test set of the OCT2017 dataset. The results of our anti-noise experiments demonstrate that our approach exhibits superior robustness compared with other methods, yielding accuracy improvements ranging from 0.6% to 2.9% when compared with ResNet under various PSNR noise conditions.

## 1. Introduction

The macula is the central area of the retina with the highest concentration of visual cells. The status of the macula may be influenced by various pathological factors, such as age-related macular degeneration (AMD) and diabetic macular edema (DME). AMD and DME are prevalent retinal diseases, affecting nearly 8% of the world’s population [1,2]. These diseases can damage patients’ vision and affect their daily lives. Proper screening tools can effectively reduce the harm caused by retinal diseases. Optical coherence tomography (OCT) [3,4,5], a novel optical imaging technology, allows for the three-dimensional imaging of human tissue microstructures and facilitates the detection of retinal disease lesions. OCT is widely used by ophthalmologists due to its non-invasive nature, low cost, and lack of side effects [6]. Identifying retinal lesions in each B-scan of the OCT volume can be time-consuming and onerous for ophthalmologists. There is also the risk of misdiagnosis or missed diagnosis. Thus, automated diagnosis of retinal disease images can assist ophthalmologists in efficiently diagnosing and treating retinal diseases.

Over the past few years, machine learning has achieved significant results in the classification of retinal disease images. Initially, preprocessing methods are applied to the input images, such as image denoising [7,8,9,10] and retinal flattening [9,11]. Subsequently, designed feature descriptors are used to extract features and then put into classifiers to acquire classification results. Albarrak et al. [12] employed preprocessing techniques to extract the ROI and eliminate noise. They combined image decomposition with LBP histograms for feature extraction and utilized the Bayesian classifier for classification. Srinivasan et al. [13] applied various preprocessing methods, such as BM3D, retinal curvature flattening, and image cropping. They then utilized multiscale histograms of gradient descriptors for feature extraction and employed SVM as the classifier. Sun et al. [14] eliminated noise and background areas via retina alignment and image cropping. They obtained image features using sparse coding and a spatial pyramid, employing SVM for classification. While machine learning methods have shown promising results, one limitation is the time-consuming and laborious task of manually extracting features. Meanwhile, moreover, the increasing number of OCT images of the retina show that traditional diagnostic methods are unable to meet the requirements of automated diagnosis.

In recent years, deep learning, specifically convolutional neural networks (CNNs), has made significant advancements in the fields of natural language processing and computer vision [15]. This success has inspired numerous researchers to employ CNNs in the field of retinal OCT images. Several deep CNN models have been applied with great success in the classification of retinal images [16,17,18,19,20]. Huang et al. [21] proposed a layer-guided convolutional neural network (LGCNN). Initially, they utilized ReLayNet to extract information about retinal layers. Subsequently, the LGCNN was employed to integrate this extracted information for the classification of retinal disorders. Fang et al. conducted two studies on retinal image classification. One study [22] adopted an iterative fusion strategy to merge features among different convolutional layers. Another study [23] proposed the lesion-aware convolutional neural network (LACNN). Their approach involves first utilizing a lesion detection network (LDN) to obtain an attention map for retinal lesions, which is then integrated into the classification framework. Sotoudeh-Paima et al. [24] first applied the feature pyramid network (FPN) [25] structure to CNNs to construct multi-scale CNN models. This model can obtain multi-scale features to perform classification well. Ma et al. [26] combined ConvNet and a transformer to extract the local and global features of OCT images, which is superior to a pure vision transformer (ViT). In addition, the transfer learning technique is widely used to classify retinal OCT images and shows promising results [18,19,20,27]. Although these classification methods achieve ideal performance, model optimization remains a significant concern for convolutional neural networks. A residual network is a classical convolutional neural network (CNN) that effectively prevents performance degradation by introducing a skip connection. Some researchers have applied ResNet to the classification of retinal diseases. Serener et al. [16] utilized transfer learning to apply ResNet18 for classifying retinal disease images, demonstrating its superiority over AlexNet. Li et al. [28] proposed a method that combines multiple ResNet50 models to enhance feature extraction. ResNet has also been selected as the backbone for multi-model retinal image classification in [29,30], providing further evidence of the effectiveness of residual networks. Karthik et al. [31] designed the “EdgeEN” block with batch normalization to replace the existing skip connection in ResNet, effectively emphasizing the retinal layers. These studies have investigated the application of residual networks to retinal classification and have demonstrated their reliability and effectiveness. However, the aforementioned deep learning methods fail to consider the impact of noise, especially speckle noise. In clinical applications, retinal images generated by OCT frequently exhibit significant speckle noise [32]. Most existing CNN models are unable to extract sufficiently discriminative features under strong noise conditions. Additionally, the size and morphology of lesion areas in OCT images of retinal diseases vary, posing a challenge for effective feature extraction. Conventional classification models that use the last convolutional layer to classify retinal images appear to be inadequate. Given the aforementioned issues, it is necessary to develop a new deep learning model for classifying retinal disease images.

Based on the above challenges, we propose a multi-scale-denoising residual network (MS-DRCN) for OCT image classification. Initially, the residual network was selected as the backbone. Considering the significant presence of speckle noise in OCT images, we developed a soft-denoising block employing soft thresholding [33,34,35], which effectively reduces the interference features during the training process. In addition, a multi-scale context block was designed to expand the receptive field and enhance the extracted features. The effectiveness of such multi-scale structures has been demonstrated in previous studies [36,37]. Retinopathy displays a wide range of variations and complexities across different scales. Inspired by the feature fusion techniques discussed in [11,22,24], we proposed a feature fusion method to obtain multi-scale information for the precise localization of lesion areas.

Specifically, the main contributions of our study are as follows:The soft-denoising block is designed to effectively remove noise-related features by employing soft thresholding.The multi-scale context block (MCB) aims to express multi-scale features and enhance feature extraction by increasing the receptive field for each subnet.The feature fusion block (FFB) is designed to combine low-resolution and high-resolution features, enabling more accurate identification of retinopathy at various scales in OCT images.Anti-noise experiments proved that the proposed method exhibits superior robustness against various intensities of speckle noise and Gaussian noise.

The subsequent sections are outlined as follows: Section 2 details the framework of our method, the dataset, and the experimental setting. Section 3 presents and analyzes the experimental results, and Section 4 draws the conclusions.

## 2. Materials and Methods

### 2.1. OCT Datasets

This section introduces two publicly available OCT datasets of the retina for subsequent experiments.

The first dataset used in our study was the publicly available dataset provided by the University of California San Diego (OCT2017) [20]. This dataset comprises 84,484 B-scans obtained from 4686 patients and includes a training set and a test set. The dataset includes four classes: Drusen, CNV, DME, and Normal. The training set consists of 83,484 retinal OCT images, with 37,205 CNV, 8616 Drusen, 11,348 DME, and 26,315 Normal images. The test set comprises 1000 retinal OCT images, with 250 images per class, obtained from a total of 633 patients.

The second dataset was named OCT-C4, which consists of 25,080 retinal OCT B-scans, including 5540 CNV, 3401 DME, 6765 Drusen, and 9374 Normal scans. This dataset consists of three sub-datasets: NEH [38], Srinivasan2014 [13], and OCT-C8 [39]. The NEH dataset, collected by Noor Eye Hospital in Tehran, Iran, comprises 12,649 retinal OCT B-scans, including 3240 CNV, 3742 Drusen, and 5667 Normal scans. The Srinivasan 2014 dataset, introduced by Duke University, comprises 723 AMD, 1101 DME, and 1407 Normal OCT images. The OCT-C8 dataset consists of eight categories, but we only used four categories (CNV: 2300, Drusen: 2300, DME: 2300, and Normal: 2300).

### 2.2. Proposed MS-DRCN Method

In this section, the MS-DRCN is proposed to classify OCT images of retinal diseases. The overall architecture of the MS-DRCN is shown in Figure 1, which consists of a residual network, subnets, and a feature fusion block (FFB). The residual network was chosen as the backbone for feature extraction. It is easy to optimize, which can effectively alleviate the problem of performance degradation by using skip connections. The subnets included a soft-denoising block (SDB) and a multi-scale context block (MCB), which can obtain multi-scale features and adaptively eliminate noise-related features. Finally, we used a feature fusion block to merge low-resolution and high-resolution features, and then the output was fed into a classifier for retinal image classification.

#### 2.2.1. Residual Learning

To improve the performance of deep learning models, it is often necessary to increase the depth of the layers. However, this can lead to issues such as gradient vanishing and gradient explosion. To address these issues, batch normalization [40] is employed during the model’s training process for optimization. When the model using batch normalization starts to converge, it may experience performance degradation issues not attributable to overfitting.

A deep residual network can effectively address these challenges. It can enhance model performance by increasing the number of convolutional layers, a benefit derived from the residual block. The key to the residual block is the introduction of a shortcut connection to facilitate the learning of the residual mapping, which is beneficial for model training and optimization. Figure 2 illustrates the residual block. Residual learning is defined as shown in Equation (1).
*y* = *f* (*x*, *w*) + *x*(1)
where *x* refers to the input feature, and *y* denotes the output feature. The function *f* (·) is the residual mapping function, while *w* represents the parameter of the residual block. The residual block consists of two convolutional layers, two BN layers, two ReLu activation functions, and an additive operation. The introduction of shortcut connections in the residual network effectively addresses the issue of performance degradation in deep neural networks.

#### 2.2.2. Soft-Denoising Block

In clinical applications, retinal images generated with OCT often suffer from speckle noise. This type of noise significantly degrades the signal-to-noise ratio and image contrast, thus posing challenges for accurately identifying retinal lesion locations. Consequently, the elimination of speckle-noise-related features has become an urgent problem that must be solved. In this section, we consider soft thresholding to effectively remove these interference features.

The soft-denoising block is shown in Figure 3. The input feature x is transformed into 1D vectors by applying global average pooling (GAP) to its absolute values. Next, the 1D vectors are passed through two fully connected (FC) layers, which are equivalent to the one proposed in [41]. The number of neurons in the FC hidden layer is set to 1/16 of the channel numbers of the input feature. Finally, a sigmoid function is applied to the output of the two FC layers.
(2)$$F(z)=\frac{1}{1+e^{-z}}$$
where *z* is the output of the two FC layers in the soft-denoising block, and *F*(*z*) denotes the scaling parameter.

Consequently, the threshold can be calculated by multiplying *F*(*z*) with the average value of |*x*|. By taking the absolute value of the input *x*, we ensure a positive threshold value. The average value of |*x*| prevents the threshold from becoming excessively large. The formula can be defined as follows:(3)τ=F(z)*average(|x|)
where *τ* represents the threshold, and *x* is the input feature map. Therefore, Equation (3) not only keeps the threshold in a reasonable range, but it is also positive.

Once the threshold has been determined, a soft threshold function is applied to remove interference information from the extracted features. Finally, the denoised features are added back to the input features. The overall process can be summarized as follows.
(4)y=φ(x,τ)+x
where *τ* is the threshold, φ(·) denotes the soft thresholding, *x* is the input, and *y* is the output.

In the past years, soft thresholding has been the core step in signal-denoising methods. Generally, the useful information is transformed into important features far away from zero, and noisy information is converted into unimportant features close to zero. Soft thresholding is used to convert the near-zero features to zero. The function of soft thresholding can be defined using Equation (5).
(5)$$y=\left\{\begin{array}{cc} x-\tau & x>\tau \\ 0 & -\tau \leq x\\ x+\tau & x<-\tau \end{array}\right.\leq \tau$$
where *x* is the input feature, *y* is the output feature, and *τ* refers to the threshold. Near-zero features are set to zeros by using soft thresholding so that noise-related features can be effectively eliminated.

#### 2.2.3. Multi-Scale Context Block

The size and morphology of lesion areas in OCT images of retinal diseases vary, making it challenging to effectively extract features using a single scale. Therefore, incorporating a multi-scale structure is necessary to enlarge the receptive field of each network layer and enhance the features.

In response to these challenges, we introduce the multi-scale context block (MCB) in this section, as depicted in Figure 4. Bconv denotes the combination of a convolution layer, a batch normalization (BN) layer, and rectified linear unit (ReLu) activation. Initially, a 1 × 1 Bconv is applied to reduce the dimensionality of the input feature channels to one-eighth of the original number. The input features are subsequently divided into four groups, with each group undergoing a 3 × 3 Bconv with a dilation rate of *k* (where *k* = 1, 2, 3, and 4). This approach enables the acquisition of multi-scale contextual information by integrating features obtained from various dilation rates. Subsequently, the features from the four groups are concatenated to aggregate the multi-scale information.
(6)$$C=Concat(F_{k}(I_{i})$$
where *I_i_* represents the feature of each group, *C* denotes the concatenated feature, and *F_k_* is the 3 × 3 convolution with a dilation rate of *k*.

The concatenated feature passes through a 1 × 1 convolution to increase the dimensionality. Finally, an element-wise summation operation is applied to integrate a skip connection into the output feature. The process can be described as follows:(7)y=x+θ(C)
where *x* is the input feature of the MCB, *θ* represents the 1 × 1 Bconv, and *C* denotes the concatenated feature. The MCB can acquire multi-scale information and enhance the extracted features.

#### 2.2.4. Feature Fusion Block

Retinopathy exhibits variety and complexity across different scales, and identifying retinal lesions based on single-scale information is ineffective. To better utilize the extracted features, a feature fusion module is introduced to acquire multi-scale information. Previous studies [11,22,24] have shown that feature fusion can significantly enhance the accuracy of retinal disease recognition. Therefore, we propose a feature fusion block that combines low-resolution and high-resolution features to obtain cross-scale information. In a CNN model, earlier feature maps have a high resolution and weak semantics, providing more detailed information, while later feature maps have a low resolution and strong semantics, containing more target feature information. By utilizing the feature fusion block (FFB) to establish a hierarchical pyramid structure, the model can capture strong semantics across all scales.

The FFB is illustrated in Figure 1. The output feature maps of the subnets vary in scales and channels. Initially, a 1 × 1 convolution is employed to adjust the channel of each scale to 256, thereby standardizing the impact of each scale. Subsequently, an upsampling operation is conducted to match the size of the high-resolution feature map with that of the low-resolution feature map. The guide attention (GA) is utilized to merge the high-resolution feature maps with the low-resolution feature maps. The FFB can be expressed as follows:(8)$$y_{i}=\left\{\begin{array}{cc} GA(x_{i},unsampled(x_{i+1})) & i=1,2,3\\ x_{i} & i=4 \end{array}\right.$$
where *x_i_* is the input of each path, *y_i_* is the output of the FFB, and GA denotes the guide attention module.

The guide attention (GA) module is shown in Figure 5. First, an upsampling operation is applied to the high-resolution feature to match the size of the low-resolution feature. Next, both features are normalized using a BN layer. Feature attention is obtained using the element-wise product operation. Consequently, a 3 × 3 BConv is employed to generate the modulated feature. Finally, the modulated feature is added to the input feature.

The GA can be expressed as:(9)$$f=\beta (x_{i})⊗\beta (upsampled(x_{i+1}))$$
(10)$$y=\varphi (f)+x_{i}$$
where *f* denotes the modulated feature, *β* (·) is the BN operation, *φ* (·) is the 3 × 3 BConv, upsampled represents the upsampling operation, *x_i_* is the low-resolution feature, and *x_i_*+_1_ is the high-resolution feature.

### 2.3. Experimental Settings

In our study, the model was trained using the OCT2017 and OCT-C4 datasets. For the OCT-C4 dataset, it was divided into a 4:1 ratio for training and testing. In each run, 20% of the training set in both datasets was selected for validation, optimizing the model for improved performance. The training process was repeated five times for each dataset, and the average of the five results was considered as the final result.

The input images were all resized to 224 × 224 for model training. Normalization was performed on the images by subtracting the mean and dividing by the standard deviation. To prevent model overfitting and improve its generalization performance, data augmentation techniques were utilized during the training process to increase the diversity of data features. These techniques included random rotation, scale changes, brightness adjustments, and horizontal flipping. The detailed information for the data augmentation strategy is shown in Table 1.

The cross-entropy loss function is commonly used to classify images. In our study, a weighted cross-entropy loss function was applied to address the problem of class imbalance in the OCT2017 and OCT-C4 datasets. Additionally, we set the number of epochs to 50 and the batch size to 32. Herein, we applied the SGD optimizer with a weight decay of 1 × 10^−4^ and momentum of 0.9 to optimize the model. The cosine annealing strategy [42] was employed to dynamically adjust the learning rate during initialization. In our study, we initialized the learning rate at 0.001 and set the final learning rate to 1 × 10^−5^. All model training was based on the transfer learning method.

The experiments were carried out on a workstation equipped with an Intel Xeon Gold 6248R CPU (Intel, Santa Clara, CA, USA) and an NVIDIA RTX A5000 GPU (NVIDIA, Santa Clara, CA, USA). We utilized the PyTorch (v. 2.1.1) deep learning framework to conduct the experiments, leveraging Cuda11.1 and CUDNN10.0 to accelerate the speed of model training.

### 2.4. Evaluation Criteria

In order to assess the effectiveness of model performance, various metrics including accuracy, precision, sensitivity, specificity, F1-score, and AUC were employed to evaluate the experimental results. These indicators are defined as follows:(11)$$Accuracy=\frac{TP+TN}{TP+FP+TN+FN}$$
(12)$$\mathrm{Pr}ecision=\frac{TP}{TP+FP}$$
(13)$$Sensitivity=\frac{TP}{TP+FN}$$
(14)$$Specificity=\frac{TN}{TN+FP}$$
(15)$$F1=\frac{2\times \mathrm{Pr}ecision\times Sensitivity}{\mathrm{Pr}ecision+Sensitivity}$$
where *TP*, *FP*, *TN*, and *FN* represent true positive, false positive, true negative, and false negative, respectively.

Based on the multi-class confusion matrix, we needed to calculate the overall accuracy (*OA*) and overall precision (*OP*) to evaluate the quantitative results. *OA* and *OP* can be defined as:(16)$$OA=\frac{1}{N}\sum_{i=1}^{N}TP_{i}$$
(17)$$OP=\frac{1}{C}\sum_{i=1}^{C}\frac{TP_{i}}{TP_{i}+FP_{i}}$$
where *N* is the number of testing samples, and *C* is the number of total categories.

## 3. Results and Discussion

### 3.1. Comparisons on OCT2017 Dataset

The quantitative results of the different methods are shown in Table 2. The methods listed in Table 2 were trained on the OCT2017 dataset using identical experimental conditions as described in Section 2.3. OA and OP refer to overall accuracy and overall precision. The results show the superior performance of the MS-DRCN model compared with well-known CNN models and the retina classification frameworks.

VGG16 [43] and ResNet [44] are the classical convolutional networks. We employed transfer learning to train them on the OCT2017 dataset. Transfer learning [20] was applied by Kermany et al. to fine-tune the Inception v3 model on the OCT2017 dataset. Therefore, the method obtained an accuracy of 94.3%. Fang et al. [22,23] proposed the IFCNN model, which adopted an iterative fusion strategy to accurately identify retinal diseases. FPN-ResNet was proposed by Sotoudeh-Paima et al. [24], who applied a feature pyramid network (FPN) structure to well-known CNN models to acquire multi-resolution information for retinal image classification. For a fair comparison, we implemented the transfer learning, IFCNN, and FPN-ResNet frameworks on the OCT2017 dataset. As shown in Table 2, our model achieved the highest performance in terms of overall accuracy (OA) and precision (OP) on the OCT2017 dataset, with values of 96.4% and 96.6%, respectively. The OA and OP exhibit an enhancement of 2.4% and 2.1% compared with ResNet. The reason may be that MS-DRCN improves the identification of lesion regions by considering multi-scale information and removing noise-related features.

Figure 6 shows the train accuracy, validation accuracy, train loss, and validation loss. In Figure 6b, we can observe that the validation loss set decreased until it reached a plateau. This suggests that there was no overfitting of the MS-DRCN during the training process. Figure 7 displays the correct (first row) and incorrect (second row) classification cases of the OCT images, along with the predicted probability scores for each category. The first row of Figure 7 demonstrates that the MS-DRCN accurately classified the retinal images with high confidence scores. The second row of Figure 7 shows the incorrect classification cases between CNV and DME. This misclassification can be attributed to their similar visual characteristics that involved significant liquid accumulation. However, there were cases where Normal images were misclassified as Drusen images with small lesions, which posed a challenge for the MS-DRCN. Moreover, the majority of misclassifications were observed between Drusen and CNV. Drusen represents the early stage of AMD, characterized by the presence of extracellular material deposits, while CNV corresponds to the later stage of AMD, characterized by the formation of neovascular membranes and significant fluid accumulation. The main reason is similarity between Drusen and CNV, which may confuse the MS-DRCN.

### 3.2. Comparisons on OCT-C4 Datasets

In order to further validate the effectiveness of the proposed method, we conducted an evaluation of MS-DRCN on the OCT-C4 dataset. The training methods and settings of MS-DRCN on the OCT-C4 dataset were consistent with those on the OCT2017 dataset. We implemented transfer learning, IFCNN, and FPN-ResNet on the OCT-C4 dataset.

Table 3 shows the quantitative analysis results of classification indicators on the OCT-C4 dataset. MS-DRCN generally outperformed other methods in terms of accuracy, precision, sensitivity, OA, and OP. Compared with ResNet, MS-DRCN showed an improvement of 2.5% and 1.7% in OA and OP. Similarly, MS-DRCN exhibited an enhancement of 1.6% and 1.6% in OA and OP compared with FPN-ResNet. The superior performance of MS-DRCN may be attributed to its soft denoising model and multi-scale structure, enabling better identification of the lesion area. As a result, MS-DRCN achieved an OA and OP of 96.5% and 96.9%.

### 3.3. Robustness of Our Method to Noise

In practical applications, original OCT images often exhibit significant speckle noise due to the interference signal caused by the backscattered light from the biological tissue [32]. Due to the characteristics of OCT imaging, the scattering noise primarily manifests as multiplicative noise. Given the substantial presence of speckle noise in OCT images, it is possible to establish a noise model that accounts for the characteristics of speckle noise. In the logarithmic domain, speckle noise can be effectively approximated as additive Gaussian noise [45]. It can be expressed as follows:(18)f=x*m+n
where *x* is the raw image never contaminated by noise, *f* is the noisy image, and *m* and *n* are multiplicative noise and additive noise, respectively. Since additive noise is very small compared with multiplicative noise, it can be ignored.

To make the classification task more challenging and evaluate the robustness of the MS-DRCN, varying intensities of additional noise were introduced into the original OCT images. Since OCT images are primarily affected by speckle noise, and Gaussian noise is commonly employed in anti-noise research [26], we introduced additional speckle noise and Gaussian noise to the test set.

The intensity of noise was determined by the value of the peak signal-to-noise ratio (PSNR). The definition of the PSNR is as follows:(19)$$\mathrm{PSNR}=20\cdot \log_{10}(\frac{Max}{MSE})$$
where *Max* denotes the maximum pixel value of the OCT image, and *MSE* refers to the mean squared error. As the value of the PSNR decreases, the severity of the added noise increases.

In our experiments, Gaussian noise and speckle noise were added to the test sets of the OCT2017 and OCT-C4 datasets, resulting in the average PSNR between the original test and its noisy version of 35, 32, 29, 26, 23, and 20 dB, respectively. Notably, we trained the models using the training set of both datasets and evaluated their performance on the test set with Gaussian or speckle noise.

Figure 8 and Figure 9 show the accuracies of the different methods on the test set of the OCT2017 and OCT-C4 datasets. It can be observed that the predicted accuracy of the MS-DRCN was better than that of the other methods under different levels of noise intensity. This means that the MS-DRCN is more robust to different levels of noise. As the PSNR decreases, speckle noise has a greater effect on image quality than Gaussian noise at the same PSNR. The reason may be that multiplicative noise has a greater impact on image quality than additive noise. The experimental results demonstrate the superior robustness of the MS-DRCN to noise.

### 3.4. Ablation Experiments

This section presents the ablation experiments to assess the effectiveness of various components, including MCB, SDB, and FFB. To evaluate our method, ResNet was selected as the baseline model, which facilitated model optimization. We gradually integrated the FFB, MCB, and SDB modules into the baseline model. After adding the new modules, we retrained the model using the OCT2017 dataset. The effectiveness of the different modules can be known through experimental comparisons.

The quantitative evaluation results on the OCT2017 dataset are shown in Table 4. It is observed that the FFB improved the overall accuracy by 0.9%, indicating the effectiveness of combining low-resolution and high-resolution features for lesion area localization. The MCB captured multi-scale information and enhanced the extracted features by integrating convolution kernels of different sizes. It improved the accuracy by 0.8% compared with the combination of ResNet and FFB. In addition, the soft-denoising block reduced noise-related features by applying soft thresholding during the training process. As a result, the MS-DRCN achieved an overall accuracy (OA), overall precision (OP), and F1-score of 96.4%, 96.6%, and 96.5%, respectively. The MS-DRCN achieved significant performance improvements with a small number of parameters compared with ResNet.

### 3.5. Visualization

Despite the remarkable success of convolutional neural networks in the field of computer vision, there is still a significant challenge regarding their interpretability. To solve this issue, gradient-weighted class activation mapping (Grad-CAM) [46] is introduced to explain the decision-making process of the CNN model. It highlights significant regions within the image that contribute to accurate predictions. Figure 10 shows the raw image and its Grad-CAM maps of CNV and DME cases. It can be observed that the original image contains a large amount of noise. The activation map of learned features (red regions in Figure 10) is valuable for accurately localizing the lesion area and determining the target type. Retinopathy exhibits a considerable degree of variability and complexity. DME typically presents as fluid-filled cysts and thickening of the retina, while CNV is characterized by retinal fluid accumulation and fibrotic scars. By integrating multi-scale information and reducing noise features, the MS-DRCN enables more accurate lesion localization and prediction.

To further understand the denoising process of the MS-DRCN model, we visualized the learned features. Figure 11 shows the raw image, output of the first residual block, and output after the SDB module in the first residual block of the MS-DRCN, respectively. The feature maps in Figure 11b show that the features exacted by the residual block contain a significant amount of noise features, which are unnecessary for image classification. Figure 11c reveals that the extracted features after soft denoising are more concentrated in the retinal region than in the background region. Clearly, the MS-DRCN is capable of reducing noise-related features during the training process, which further demonstrates the effectiveness of our proposed method.

## 4. Conclusions

This study introduced a multi-scale denoising residual convolutional network (MS-DRCN) for classifying OCT images of retinal diseases. The key contributions of this research are summarized as follows: (a) designing a soft-denoising block that gradually reduces noise features while preserving useful information; (b) designing a multi-scale context block to capture multi-scale information and enhance extracted features by employing convolutions with varying dilations; and (c) the application of a feature fusion module to integrate features from both low and high resolutions. Distinguishing itself from well-known CNN models and existing multi-scale frameworks, the proposed method not only exploits cross-scale complementarity but also utilizes multi-scale information on a single scale. Furthermore, the impact of speckle noise is effectively mitigated by integrating a soft-denoising block into the MS-DRCN.

The results indicate that the MS-DRCN outperforms well-known models and several recently proposed frameworks for retinal OCT classification. Specifically, the MS-DRCN achieves an accuracy improvement of 2.4% and 2.5% over ResNet50 on the OCT2017 dataset and the OCT-C4 dataset, respectively. Anti-experiments also showed that the MS-DRCN performs better than other methods against various types of noise on the OCT2017 dataset. These results suggest that our method exhibits a superior generalization ability and robustness.

## Figures and Tables

**Figure 1 sensors-24-00150-f001:**
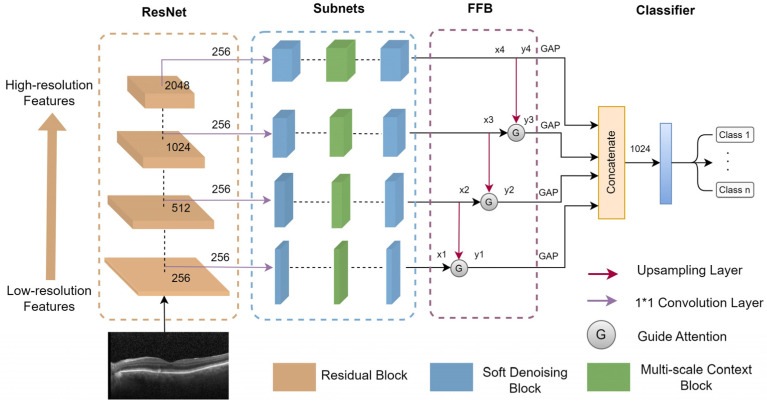
The overall architecture of MS-DRCN.

**Figure 2 sensors-24-00150-f002:**
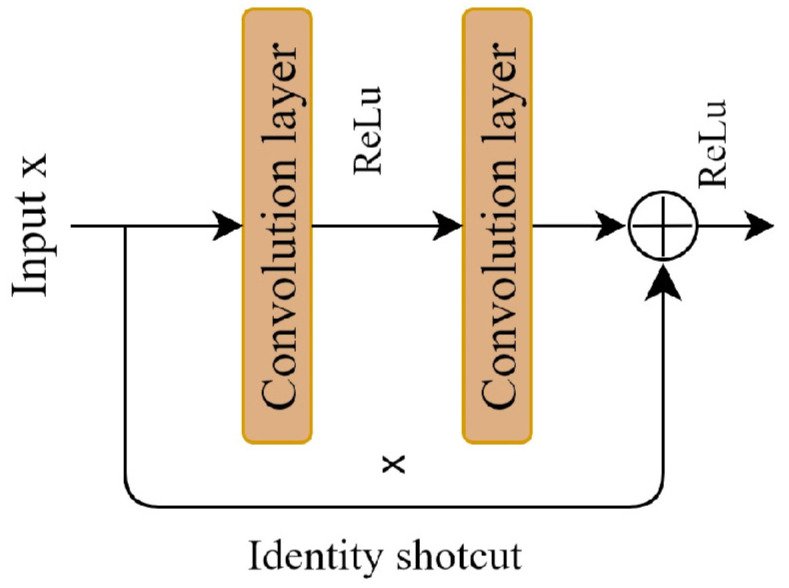
Residual learning block.

**Figure 3 sensors-24-00150-f003:**
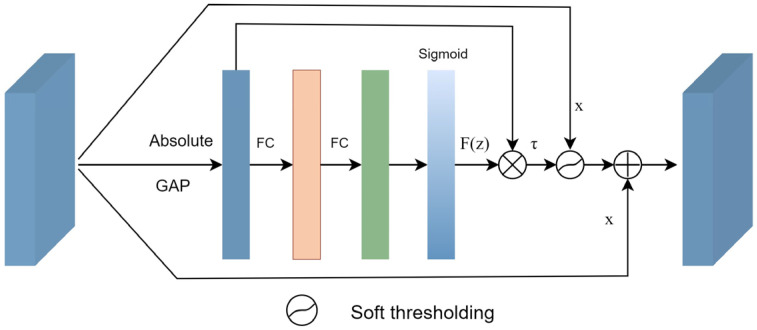
Soft-denoising block.

**Figure 4 sensors-24-00150-f004:**
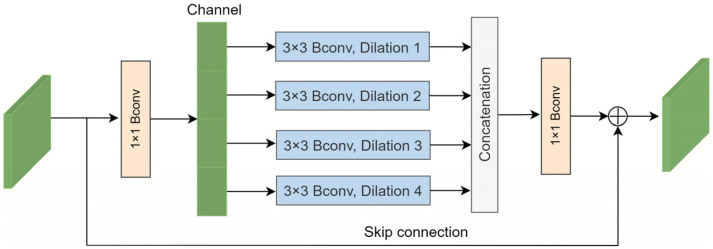
Multi-scale context block.

**Figure 5 sensors-24-00150-f005:**
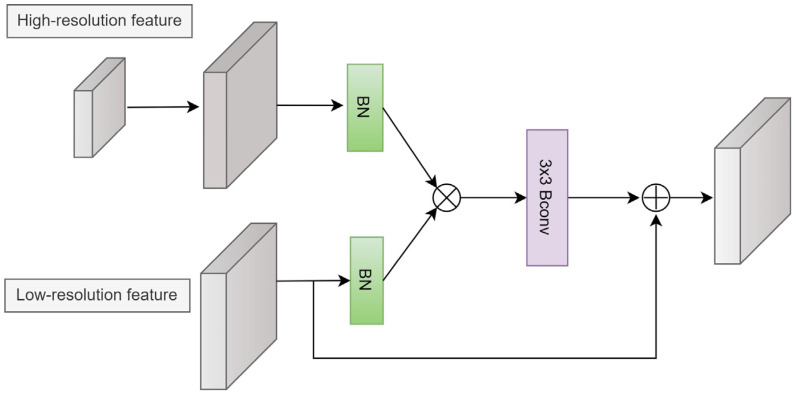
Guide attention module.

**Figure 6 sensors-24-00150-f006:**
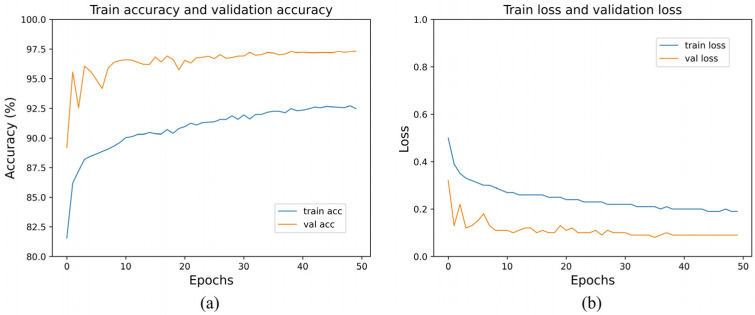
(**a**) Training accuracy and validation accuracy of MS-DRCN on the OCT2017 dataset; (**b**) training loss and validation loss on the OCT2017 dataset.

**Figure 7 sensors-24-00150-f007:**
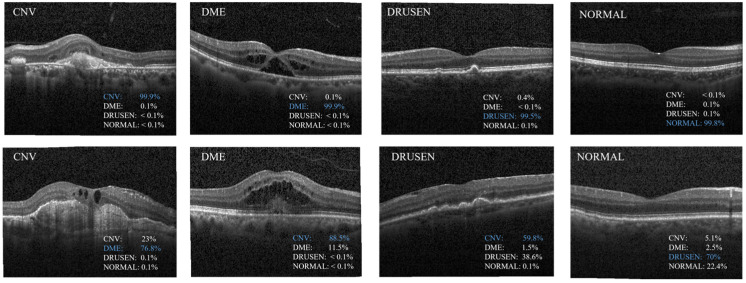
Examples of classification results predicted by the proposed MS-DRCN. The first row shows the cases of correct classification; the second row shows the cases of misclassification.

**Figure 8 sensors-24-00150-f008:**
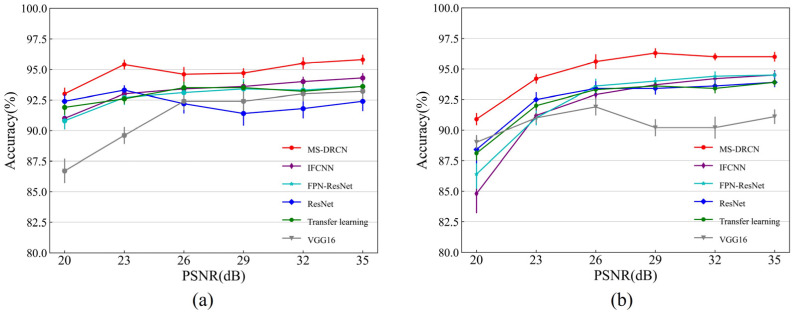
The accuracy of different methods on the test set of OCT2017 dataset with (**a**) Gaussian noise and (**b**) speckle noise.

**Figure 9 sensors-24-00150-f009:**
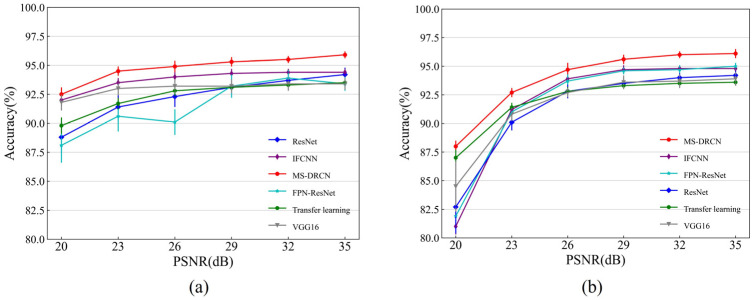
The accuracy of different methods on the test set of OCT-C4 dataset with (**a**) Gaussian noise and (**b**) speckle noise.

**Figure 10 sensors-24-00150-f010:**
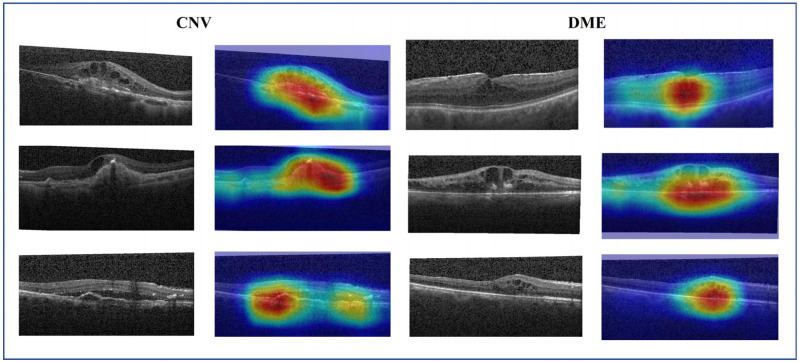
The raw image and its Grad-CAM of the CNV and DME cases. The red and yellow regions represent the area of interest to the proposed model.

**Figure 11 sensors-24-00150-f011:**
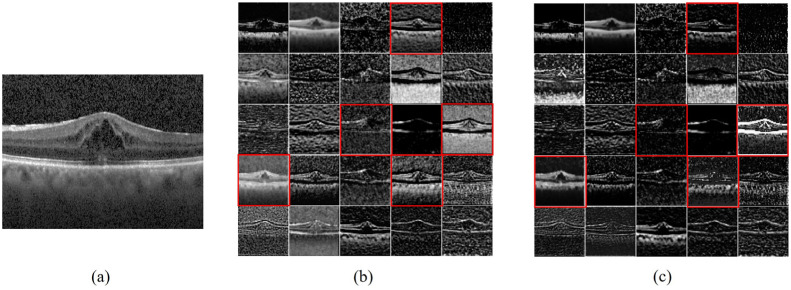
Feature maps visualization. (**a**) Raw image. (**b**) The output of first residual block of MS-DRCN. (**c**) The output after SDB module in the first residual block of MS-DRCN. The feature maps presented in (**c**) exhibit a higher concentration on the retinal area, specifically in the ones marked by the red boxes, in contrast with (**b**).

**Table 1 sensors-24-00150-t001:** Augmentation strategy in our study.

Augmentation Type	Value
Rotation range	±15 degrees
Scale range	±20%
Brightness range	±20%
Horizontal flip	True

**Table 2 sensors-24-00150-t002:** Comparison results for different classification methods on the OCT2017 dataset.

Method	Class	Accuracy (%)	Precision (%)	Sensitivity (%)	OA (%)	OP (%)
Vgg16 [43]	CNV	94.6 ± 0.4	82.5 ± 0.6	99.1 ± 0.2	93.4 ± 0.5	94.3 ± 0.5
	DME	98.9 ± 0.3	96.6 ± 1.6	98.3 ± 0.3		
	Drusen	94.6 ± 0.5	99.3 ± 0.2	78.6 ± 1.2		
	Normal	98.9 ± 0.2	98.5 ± 0.2	97.0 ± 1.0		
ResNet [44]	CNV	95.0 ± 0.5	85.0 ± 1.4	97.2 ± 1.2	94.0 ± 0.6	94.5 ± 0.5
	DME	98.7 ± 0.4	95.5 ± 1.5	99.5 ± 0.2		
	Drusen	95.1 ± 0.6	99.3 ± 0.2	81.0 ± 2.7		
	Normal	98.9 ± 0.2	98.2 ± 0.6	98.1 ± 1.0		
Transfer learning [20]	CNV	95.0 ± 0.7	85.4 ± 2.1	98.9 ± 0.5	94.3 ± 0.8	94.8 ± 0.6
	DME	99.1 ± 0.4	98.4 ± 1.3	97.8 ± 0.6		
	Drusen	95.2 ± 0.6	97.9 ± 0.2	82.6 ± 2.4		
	Normal	98.8 ± 0.2	97.3 ± 0.4	98.0 ± 0.9		
IFCNN [22]	CNV	95.4 ± 0.3	84.8 ± 0.7	99.0 ± 0.5	94.7 ± 0.2	95.2 ± 0.3
	DME	98.8 ± 0.4	98.4 ± 0.2	95.5 ± 0.3		
	Drusen	96.0 ± 0.4	98.0 ± 0.4	85.1 ± 0.8		
	Normal	98.7 ± 0.3	98.4 ± 0.3	98.5 ± 0.4		
FPN-ResNet [24]	CNV	95.9 ± 0.3	87.2 ± 0.6	98.2 ± 0.5	94.6 ± 0.2	94.9 ± 0.2
	DME	98.8 ± 0.1	96.1 ± 0.4	99.2 ± 0.2		
	Drusen	95.7 ± 0.2	98.8 ± 0.2	83.4 ± 0.8		
	Normal	98.9 ± 0.2	97.8 ± 0.2	97.3 ± 0.4		
MS-DRCN	CNV	97.4 ± 0.3	91.1 ± 1.0	99.2 ± 0.2	96.4 ± 0.2	96.6 ± 0.2
	DME	99.3 ± 0.3	98.5 ± 1.0	98.8 ± 0.1		
	Drusen	97.1 ± 0.2	97.9 ± 0.2	90.1 ± 0.7		
	Normal	99.0 ± 0.2	98.7 ± 0.2	97.5 ± 0.4		

**Table 3 sensors-24-00150-t003:** Comparison results for different classification methods on the OCT-C4 dataset.

Method	Class	Accuracy (%)	Precision (%)	Sensitivity (%)	OA (%)	OP (%)
Vgg16 [43]	CNV	97.5 ± 0.5	97.5 ± 1.1	91.4 ± 1.4	93.5 ± 0.6	94.3 ± 0.3
	DME	99.2 ± 0.2	96.2 ± 0.3	97.6 ± 0.5		
	Drusen	94.5 ± 0.5	91.8 ± 2.1	87.5 ± 0.8		
	Normal	95.8 ± 0.3	91.7 ± 1.2	97.7 ± 0.5		
ResNet [44]	CNV	98.1 ± 0.2	97.2 ± 0.6	94.1 ± 1.2	94.0 ± 0.4	95.2 ± 0.4
	DME	99.3 ± 0.2	98.4 ± 0.5	97.0 ± 0.4		
	Drusen	95.0 ± 0.5	94.0 ± 0.7	87.0 ± 1.1		
	Normal	96.0 ± 0.5	91.4 ± 1.4	98.5 ± 0.6		
Transfer learning [20]	CNV	98.0 ± 0.2	96.7 ± 0.3	94.1 ± 0.6	93.7 ± 0.3	94.6 ± 0.2
	DME	99.3 ± 0.3	97.3 ± 1.3	97.9 ± 0.6		
	Drusen	94.4 ± 0.3	94.4 ± 0.5	84.2 ± 1.1		
	Normal	95.6 ± 0.3	90.4 ± 0.9	98.6 ± 0.4		
IFCNN [23]	CNV	98.1 ± 0.2	97.3 ± 0.2	94.5 ± 0.5	94.9 ± 0.3	95.5 ± 0.3
	DME	99.2 ± 0.2	97.0 ± 0.5	97.4 ± 0.4		
	Drusen	95.7 ± 0.2	94.1 ± 0.2	89.7 ± 0.5		
	Normal	96.7 ± 0.2	93.4 ± 0.3	98.0 ± 0.2		
FPN-ResNet [24]	CNV	98.4 ± 0.2	97.0 ± 0.4	95.8 ± 0.7	94.9 ± 0.4	95.3 ± 0.5
	DME	99.2 ± 0.2	96.0 ± 1.9	98.5 ± 0.3		
	Drusen	95.7 ± 0.3	95.1 ± 0.7	88.7 ± 0.8		
	Normal	96.5 ± 0.3	93.2 ± 0.2	97.6 ± 0.8		
MS-DRCN	CNV	98.7 ± 0.2	97.4 ± 0.3	96.8 ± 0.4	96.5 ± 0.3	96.9 ± 0.4
	DME	99.6 ± 0.1	98.4 ± 0.2	98.7 ± 0.3		
	Drusen	96.9 ± 0.2	96.4 ± 0.6	91.9 ± 0.6		
	Normal	97.0 ± 0.2	95.3 ± 0.3	98.7 ± 0.3		

**Table 4 sensors-24-00150-t004:** Comparison results for different classification methods on the OCT-2017 dataset.

Method	OA (%)	OP (%)	F1 (%)	Parameter
ResNet	94.0 ± 0.6	94.5 ± 0.5	94.2 ± 0.2	24.5 M
ResNet + FFB	94.9 ± 0.4	95.2 ± 0.3	95.1 ± 0.4	26.7 M
ResNet + FFB + MCB	95.7 ± 0.2	95.9 ± 0.2	95.8 ± 0.2	26.7 M
MS-DRCN	96.4 ± 0.2	96.6 ± 0.2	96.5 ± 0.2	27.3 M

## Data Availability

The dataset used in this study is available in Section 2.1.

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
