# Peer review of "Multi-Scale-Denoising Residual Convolutional Network for Retinal Disease Classification Using OCT"

_sensors, 2023, doi:10.3390/s24010150_

Round 1
Reviewer 1 Report
Comments and Suggestions for Authors
Macular pathologies can cause significant vision loss. Optical coherence tomography (OCT) images of the retina can assist ophthalmologists in diagnosing macular diseases. Traditional deep learning networks for retinal disease classification cannot extract discriminative features under strong noise conditions in OCT images. In this manuscript, the authors have proposed a multi-scale denoising residual convolutional network (MS-DRCN) for classifying retinal diseases to address this issue. Specifically, MS-DRCN includes a soft denoising block (SDB), multi-scale context block (MCB), and feature fusion block (FFB). SDB can automatically determine the threshold for soft thresholding, which efficiently removes speckle noise features. MCB has been designed to capture multi-scale context information and strengthen extracted features. FFB is dedicated to integrating high-resolution and low-resolution features to identify variable lesion areas precisely. The proposed approach achieved classification accuracies of 96.4% and 96.5% on the OCT2017 and OCT-C4 public datasets, respectively, outperforming other classification methods. To evaluate the robustness of the proposed approach, the authors introduced Gaussian noise and speckle noise with varying PSNRs into the test set of the OCT2017 dataset. The results of the anti-noise experiments demonstrate that the approach proposed by the authors in this study exhibits superior robustness compared to other methods, yielding accuracy improvements ranging from 0.6% to 2.9% when compared to ResNet under various PSNR noise conditions.
I have found the study quite interesting. In terms of accuracies, there have been some improvements over the other studies. I would advise the incorporation of the following suggestions before the manuscript can be accepted.
1. Authors should carefully proofread the manuscript. There are several spelling errors, e.g., “information.”
2. In the introduction section, a thorough discussion is needed on why the multi-scale denoising residual convolutional network (MS-DRCN) was used for classifying retinal diseases. The discussion should be supported by the existing literature and not just the hypothesis of the authors.
3. Why only speckle noise was added is not clear. A thorough discussion is needed.
4. All the accuracies are mentioned as mean +/- SD. Then, why Fig.8 has been plotted with mean data? The authors must include SD for each of the points. This will give a better understanding of the results.
5. In Figure 10, the description of the outputs is not self-explanatory. The authors must include sufficient details in the figure/caption so that the readers can understand the result easily.
Comments on the Quality of English Language
Minor editing of English language required
Reviewer 2 Report
Comments and Suggestions for Authors
The authors proposed a multi-scale denoising residual convolutional network (MS-DRCN) for classifying retinal diseases that includes soft denoising block (SDB), multi-scale context block (MCB), and feature fusion block (FFB). SDB can automatically determine the threshold for soft thresholding, and MCB is used capturing multi-scale context information and strengthen extracted features. FFB integrates HR and LR features identifying variable lesion areas. Novel approach appears to obtain classification accuracies of 96.4% and 96.5% on the OCT2017 and OCT-C4 datasets, respectively, outperforming SOTA classification methods. The authors evaluated the robustness of their method in presence of G additive and speckle noises into the test set of the OCT2017 dataset justifying superior robustness of novel approach against SOTA methods.
Comments:
1) The authors used different parameters during training process. For example, they chosen number of epochs to 50, the batch size of 32. They employed SGD optimizer with weight decay of 1e-4 313 and momentum of 0.9 to optimize the model. This reviewer considers that authors should explain the reason to select these parameters values.
2) This reviewer thinks that for better understanding of potential reader it would be better presenting performance curves for training and validation processes for loss function o/y accuracy as functions of epoch number.
3) In opinion of this reviewer, the authors should explain the conditions for comparing the performance of their framework against different systems from SOTA presented in tables 2, 3 for OCT2017 and OCT-C4 datasets. The authors should clarify if the values of criteria (ACC, PRE, SEN) have been used from mentioned papers for these databases or the authors of this study performed their experiments employing codes of such frameworks. For example, analyzing the framework IFCNN [36], the reviewer found for this system (see table 1 in 36) that the results for mentioned framework have been obtained for UCSD dataset with 10-fold cross-validation method. So, their results are found in different experimental setup that this reviewing study used.
4) In subsect. 3.3, the authors explained robustness of their method performing experimental results for noisy images when they contaminated dataset images according to model (18) introducing speckle and additive noises. The PSNR criterion depends on intensity of speckle noise as well as additive one. Some additional explications should be provided in this part.
Round 2
Reviewer 2 Report
Comments and Suggestions for Authors
The authors have attended all comments of this reviewer.